# Prenatal Exposures, Diagnostic Outcomes, and Life Experiences of Children and Youths with Fetal Alcohol Spectrum Disorder

**DOI:** 10.3390/nu16111655

**Published:** 2024-05-28

**Authors:** Svetlana Popova, Danijela Dozet, Mary-Rose Faulkner, Lesley Howie, Valerie Temple

**Affiliations:** 1Institute for Mental Health Policy Research, Centre for Addiction and Mental Health, 33 Ursula Franklin Street, Toronto, ON M5S 2S1, Canada; danijela.dozet@camh.ca (D.D.); maryrose.faulkner@mail.utoronto.ca (M.-R.F.); 2Institute of Medical Science, Faculty of Medicine, University of Toronto, Medical Sciences Building, 1 King’s College Circle, Toronto, ON M5S 1A8, Canada; 3Dalla Lana School of Public Health, University of Toronto, 155 College Street, Toronto, ON M5T 3M7, Canada; 4Factor-Inwentash Faculty of Social Work, University of Toronto, 246 Bloor Street W, Toronto, ON M5S 1V4, Canada; 5North Island Hospital Comox Valley, 101 Lerwick Rd, Courtenay, BC V9N 0B9, Canada; lesley.howie@islandhealth.ca; 6Surrey Place, 2 Surrey Place, Toronto, ON M5S 2C2, Canada; valerie.temple@surreyplace.ca

**Keywords:** fetal alcohol spectrum disorder, fetal alcohol syndrome, diagnosis, prenatal alcohol exposure, prenatal drug exposure, morbidity

## Abstract

Children and youths diagnosed with FASD may experience a range of adverse health and social outcomes. This cross-sectional study investigated the characteristics and outcomes of children and youths diagnosed with FASD between 2015 and 2018 at the Sunny Hill Centre in British Columbia, Canada and examined the relationships between prenatal substance exposures, FASD diagnostic categories, and adverse health and social outcomes. Patient chart data were obtained for 1187 children and youths diagnosed with FASD and analyzed. The patients (mean age: 9.7 years; range: 2–19) had up to 6 physical and 11 mental health disorders. Prenatal exposure to other substances (in addition to alcohol) significantly increased the severity of FASD diagnosis (OR: 1.18): the odds of FASD with sentinel facial features (SFF) were 41% higher with prenatal cigarette/nicotine/tobacco exposure; 75% higher with exposure to cocaine/crack; and two times higher with exposure to opioids. Maternal mental health issues and poor nutrition also increase the severity of FASD diagnosis (60% and 6%, respectively). Prenatal exposure to other substances in addition to alcohol significantly predicts involvement in the child welfare system (OR: 1.52) and current substance use when adjusted for age (aOR: 1.51). Diagnosis of FASD with SFF is associated with an increased number of physical (R^2^ = 0.071, F (3,1183) = 30.51, *p* = 0.000) and mental health comorbidities (R^2^ = 0.023, F (3,1185) = 9.51, *p* = 0.000) as compared to FASD without SFF adjusted for age and the number of prenatal substances. Screening of pregnant women for alcohol and other substance use, mental health status, and nutrition is extremely important.

## 1. Introduction

Alcohol is a well-known teratogen that can lead to a range of adverse pregnancy and neonatal outcomes, including stillbirths, intrauterine growth restriction, small-for-gestational-age infants, spontaneous abortions, and pre-term births [1,2,3,4,5]. Prenatal alcohol exposure (PAE) causes central nervous system damage to the developing fetus and has been shown to affect all organ systems of the fetus [6]. Examples of these teratogenic effects include impacted brain functioning, immune system alterations [7], impacted emotion regulation [8], organ defects [9], and impacted gastrointestinal and endocrine systems of the child. An estimated 1 in every 13 prenatally exposed children will develop fetal alcohol spectrum disorder [10], a chronic neurodevelopmental disorder that affects approximately 2–3% of the general population in Canada [11], translating into a conservative estimate of over 976,153 people with FASD across the country, based on the current population size [12].

Children with FASD may experience deficits across the cognitive, physical, emotional, and behavioral domains [13,14]. As a result, they typically have comorbid physical and mental health conditions that require greater utilization of healthcare and community services, including support services in areas such as learning, memory, attention, communication, emotional regulation, and adaptive daily living skills [15,16]. This is evident in the high cost burden associated with FASD in the Canadian healthcare systems, which has been estimated at CAD 1.8 billion annually [17]. Each individual with FASD is unique and has areas of both strengths and challenges [18], wherein support services can facilitate healthier lives where they can show their strengths and work to fulfill their unique potential.

In addition to the physical and mental health comorbidities, many individuals with FASD also experience adverse family and social outcomes throughout their lifespan. In childhood, individuals with FASD have a higher likelihood of having adverse childhood experiences (ACEs) [19], which also increases their risk of developing additional comorbid neurodevelopmental disorders [20]. Children with FASD are also more likely to be exposed to substance use in their household, to have difficulties in school and/or employment, to be victims and/or perpetrators of crime, and to be involved with the child welfare, foster care, and correctional systems [19,21,22,23]. Being born in unfavorable circumstances related to their familial or social situations is compounded by these personal life experiences, which impact not only the individual with FASD, but also the family and surrounding community.

It has been widely accepted that alcohol causes the most adverse neurobehavioral effects on a fetus, when compared to cannabis, cocaine, and heroin [24]; however, exposure to substances in addition to alcohol, namely tobacco and cannabis, can act synergistically to increase the risk of adverse neonatal and childhood outcomes for individuals with FASD [25]. In one Canadian study, an estimated 68% of children with suspected FASD were also prenatally exposed to tobacco and/or cannabis [11]. Prenatal exposure to other substances, including opioids, cocaine, and amphetamines, can also cause conditions such as neonatal abstinence syndrome or neonatal opioid withdrawal syndrome, which can increase the complexity of neonatal care services that may be required by children with FASD. In addition, alcohol-exposed pregnancies are more likely to involve mothers with a history of mental health issues and substance use, and are also more likely to be an unplanned pregnancy. These factors are further exacerbated by the fact that alcohol-exposed pregnancies tend to receive delayed prenatal care [26] and fewer antenatal visits, which limits the potential for prenatal care providers to screen for substance use and facilitate access to brief interventions that could potentially aid in harm reduction.

In Canada, the majority of individuals with FASD are undiagnosed [27,28] and this has impeded research focusing on understanding the unique experiences and needs of individuals with this disorder, as well as the ability to provide appropriate services. Receiving an FASD diagnosis is key to facilitating access to FASD-informed support that can assist individuals in reducing adverse outcomes and improve their quality of life. Using clinical chart data, this study gathered information regarding the health and social outcomes of children and youths aged 2–19 years with diagnosed FASD (2015–2018) within a region of British Columbia (BC), Canada. The study aimed to (1) examine the demographics, prenatal exposures, comorbidities, and child welfare/criminal justice system involvement of the children and youths; (2) explore the associations between various prenatal substance exposures and FASD diagnostic categories; (3) investigate the associations between FASD diagnostic categories and adverse health and social outcomes; and (4) investigate the associations between prenatal substance exposures and adverse health and social outcomes.

## 2. Materials and Methods

### 2.1. Study Design and Data Source

This cross-sectional, clinic-based study used patient chart data housed at the Sunny Hill Centre for Children in BC, Canada. This Centre is part of the Complex Developmental Behavioural Conditions (CDBC) network, which provides diagnostic assessments across the province for children and youths aged 18 months to 19 years who have been prenatally exposed to substances or are suspected to have an intellectual disorder. These services are delivered by region through local health authorities and housed at multiple locations. The Centre serves the Vancouver Coastal and Fraser Health Authority regions, which include large urban communities such as the greater Vancouver region, the most populous city in BC, and more rural and remote settings outside of major centers. Collectively, the two regions where the Centre provides diagnostic assessments have a population of over 3 million people.

### 2.2. Study Population

The study population consisted of children and youths (aged 2–19 years) who accessed the Centre’s CDBC program and were diagnosed with FASD between 2015 and 2018. Data were abstracted retrospectively in 2020 from patient charts based on information available at the time of the assessment and inserted into a spreadsheet of pre-outlined variables. These consisted of (1) demographic information, including age, sex, year of diagnosis, FASD diagnosis, living arrangement, geographic location, languages spoken at home, adoption status, past personal/sibling involvement in the child welfare system, and past personal involvement in the legal justice system; and (2) medical information, including prenatal exposures and mental health/physical health comorbidities. Child welfare system involvement constituted the involvement of the Ministry of Children and Family Development for the child/youth diagnosed with FASD or their siblings, as well as assistance given to the parents. Data on prenatal exposures and maternal/neonatal conditions included both confirmed and suspected exposures and were not mutually exclusive categories. The physical and mental health comorbidity categories denoted confirmed comorbidities that were either indicated or newly detected within the assessment (mutually non-exclusive).

### 2.3. Statistical Analyses

A cross-sectional analysis of the patient data from the provisioned dataset was performed. Descriptive statistics were generated for the quantitative data on patient demographics, prenatal exposures, substance use, and physical and mental health comorbidities using Microsoft Excel 2017, R version 3.6.3, using RStudio version 1.1.463. For variables wherein free-text responses were initially provided, the qualitative data were converted to quantitative data using thematic analysis by listing all responses verbatim and amalgamating similar responses together. The comorbidity categories were amalgamated based on their similarity and ICD-10 or DSM codes, where applicable, and coding was completed in adherence to the new categories. Cell counts with frequencies of fewer than five were suppressed to protect patient privacy and confidentiality.

In the patient charts, FASD diagnoses were recorded using two Canadian guidelines (Chudley et al., 2005 [29] and Cook et al., 2016 [30]) to investigate the associations between (1) prenatal exposures and FASD diagnostic outcome; (2) FASD diagnostic outcome and health/social outcomes; and (3) prenatal substance exposures in addition to alcohol and health/social outcomes. Patients in the study were assigned to one of two diagnostic categories as per the newest Canadian diagnostic guidelines [30]: “FASD with sentinel facial features” (included FAS) and “FASD without sentinel facial features” (included alcohol-related neurodevelopmental disorder (ARND) and partial FAS (pFAS)).

Logistic and linear regressions were performed with the significance level set to 0.05 alpha. Odds ratios for the logistic regression models and R^2^ and F values for the linear regression models were produced for the FASD diagnostic outcomes based on prenatal exposures. Odds ratios and linear regressions for the health/social outcomes based on FASD diagnoses were produced using (a) crude estimates, (b) estimates adjusted for patient age, and (c) estimates adjusted for patient age and the number of prenatal substances. Odds ratios for health/social outcomes based on prenatal exposures in addition to alcohol were produced using (a) crude estimates, (b) estimates adjusted for patient age, and (c) estimates adjusted for patient age and the FASD diagnostic category.

## 3. Results

### 3.1. Patient Demographics

In total, 1187 children and youths were diagnosed with FASD in the Centre’s program from 2015 to 2018. About 61% (*n* = 729) were males. The age at diagnosis ranged from 2 to 19 years, with an average of 9.7 years (SD = 3.9 years). About 53% (*n* = 628) were aged from six to ten years at the time of diagnosis. Of the children and youths diagnosed with FASD, 85% (*n* = 967, 81.5%) received one of the diagnoses without sentinel facial features (FASD w/o SFF; ARND; or pFAS). Within this timeframe, 365 (30.7%) received a diagnosis in 2015, 273 (23.0%) received a diagnosis in 2016, 301 (25.4%) received a diagnosis in 2017, and 248 (20.9%) received a diagnosis in 2018.

For ninety-eight percent (*n* = 1163) of the children and youths, the primary language spoken at home was English only. At the time of the assessment, thirty-six percent (*n* = 428) resided with their biological parent(s), 25% (*n* = 303) lived in foster care, 18% (*n* = 218) lived with extended family, and about 16% (*n* = 188) lived in an adoptive home. About 17% (*n* = 200) of the children and youths were adopted and 76% (*n* = 903) lived in an urban area. Approximately 13% (*n* = 151) of the children and youths themselves or their siblings were involved in the child welfare system and 1.4% (*n* = 17) were involved in the justice system as a “perpetrator”.

About 14% (*n* = 161) of the children and youths diagnosed with FASD had at least one sibling with FASD and 2.9% (*n* = 34) had a sibling (or siblings) with possible/suspected FASD/PAE. About 15% (*n* = 174) of the children and youths diagnosed with FASD had a parent (or parents) with suspected/confirmed FASD/PAE (please see Table 1).

### 3.2. Maternal Conditions

The three most prevalent maternal conditions during the prenatal period of the children and youths diagnosed with FASD were as follows: 11.4% (*n* = 135) underwent a surgical procedure, had an injury, had a physical health issue (e.g., gestational diabetes), and/or had an undefined medical condition; 9.4% (*n* = 112) experienced pregnancy complications; and 6.8% (*n* = 81) received no/poor/limited/irregular/delayed prenatal care (see Table 2). Patient charts indicated stress during pregnancy for 4.9% (*n* = 58) of mothers, with 3.4% (*n* = 40) having a reported/suspected mental disorder during pregnancy, and 4.5% (*n* = 54) of the mothers were living in abusive environments during their pregnancy.

### 3.3. Prenatal Substance Exposure

Data on prenatal exposures are based on the information that was available in the child’s records, and not all women were screened for every substance. On average, the children and youths diagnosed with FASD were exposed to 1.2 substances prenatally, in addition to alcohol. In addition to alcohol, the following prenatal exposures were common: 33.4% (*n* = 396): cigarettes/tobacco/nicotine; 27.0% (*n* = 321): cocaine/crack; and 24.3%: (*n* = 288) marijuana. Over ten percent (*n* = 121) of the children and youths were prenatally exposed to opioids (excluding children and youths of mothers who had received methadone maintenance treatment). An estimated 0.4% (*n* = 5) of the children and youths were born to mothers who received methadone maintenance treatment during their pregnancy, and 0.5% (*n* = 6) of the children and youths experienced polysubstance exposure, which was reported by the mother. Fewer than five of the children and youths were born to mothers who had an admission to rehabilitation or a clinic due to substance use at some point during their respective pregnancy (timing unknown) (Table 3).

### 3.4. Physical Comorbidities

The five most prevalent physical comorbidities of the children and youths diagnosed with FASD were as follows: 10.1% (*n* = 120) had diseases of the respiratory system, 9.4% (*n* = 112) had diseases of the nervous system, 9.0% (*n* = 107) had genetic/congenital deformations/malformations, 7.3% (*n* = 87) had eye-related comorbidities, and 4.8% (*n* = 57) had ear-related comorbidities (Table 4). The number of physical comorbidities ranged from 0 to 6, with an average of 0.68 (SD: 1.02). A statistically significant relationship was found between age and the number of physical health comorbidities (R^2^ = 0.0046, F (1,1185) = 5.45, *p* = 0.0197).

The five most prevalent mental health problems were ADHD (66%; *n* = 789); specific learning disorders (reading, writing, mathematics, scholastic skills) (42%; *n* = 497); language/speech delays or disorders (34%); intellectual disabilities (mild/moderate or unspecified) (27%, *n* = 400); and anxiety or panic disorders (21%, *n* = 254).

The number of mental health problems ranged from 0 to 11, with an average of 2.42 (SD: 1.31). A statistically significant relationship was found between age and the number of mental health comorbidities (R^2^ = 0.005, F (1,1185) = 6.53, *p* = 0.0108).

Seventy-four children and youths with FASD (6.2% out of 1187) reported current substance use (not presented in the table), the majority (97%) of which were 13 years of age or older. Thirteen were between the ages of 11 and 15 (17.6%), and 59 were aged 16 or older (80%). Among these individuals, the most commonly used substances reported were marijuana (66.2%; *n* = 49), alcohol (48.6%; *n* = 36), and stimulants (20.3%; *n* = 15). There were 5 individuals (6.8%) with an indication of polysubstance use, fewer than 5 individuals with an indication of overdose, and 12 individuals (16.2%) who were treated for substance use. As expected, age significantly predicted substance use, with 1.88-times-higher odds of substance use for older individuals (95% CI: 1.67, 2.13; *p* = 0.000).

To measure the effects of prenatal exposures on FASD diagnostic outcomes, the children and youths were grouped into two categories: FASD with sentinel facial features (SFF: 11.9%; *n* = 141) and FASD without SFF (88.1%; *n* = 1046). FASD without SFF was set as the reference group. Logistic regression analyses found that the odds of FASD with SFF were 41% higher with prenatal cigarette/nicotine/tobacco exposure; 75% higher with cocaine/crack exposure; and 201% higher with exposure to opioids (see Table 5). Prenatal exposure to marijuana did not seem to be associated with increased odds of being diagnosed with FASD with SFF. Maternal mental health issues and poor nutrition increased the odds of FASD with SFF diagnosis as well (60% higher and 6% higher, respectively). Lastly, there was a significant association between the number of prenatal substances in addition to alcohol and the odds of the child having FASD with SFF (OR = 1.18 (1.04, 1.33), *p* = 0.009).

Table 6 presents the odds ratios and linear regression between the health and social outcomes based on the diagnoses of FASD with SFF (11.9%; *n* = 141) and FASD without SFF (88.1%; *n* = 1046). The logistic regression analyses found that diagnosis of FASD with SFF was associated with lower odds of child welfare (aOR = 0.87) and criminal justice system (aOR = 0.78) involvement, which were not statistically significant when adjusted for age and the number of prenatal substances. The crude odds ratios show that FASD with SFF was associated with a decreased odds of current substance use, including when adjusted for age and the number of substances the individual was exposed to prenatally. Linear regressions revealed statistically significant associations between FASD with SFF and the number of physical and mental health comorbidities when adjusted for age and the number of prenatal substances the child or youth was exposed to.

Table 7 presents the logistic regression models predicting adverse health and social outcomes based on the number of prenatal substance exposures in addition to alcohol exposure. The number of prenatal substances (in addition to alcohol) was associated with an increased odds of child welfare system involvement (OR: 1.52, 95% CI: 1.35, 1.71, *p* = 0.000), including when adjusted for age and the FASD diagnostic category. The number of prenatal substances (in addition to alcohol) was associated with a slightly higher odds of justice system involvement (OR: 1.33, 95% CI: 0.94, 1.88, *p* = 0.106) when adjusted for age and the FASD diagnostic category, though this was not statistically significant. Lastly, the number of prenatal substances (in addition to alcohol) significantly predicted current substance use when adjusted for age and age + the FASD diagnostic category (aOR: 1.51, 95% CI: 1.18, 1.93, *p* = 0.001).

## 4. Discussion

This study investigated certain characteristics and outcomes for 1187 children and youths diagnosed with FASD in a province of Canada and examined the relationships between their FASD diagnostic category, prenatal exposures in addition to alcohol, and adverse health and social outcomes. FASD with SFF, a diagnosis typically indicating a greater severity of alcohol effects, was significantly more likely to be given following prenatal exposure to alcohol and crack/cocaine or alcohol and opioids. Receiving a diagnosis of FASD with SFF was, in turn, associated with higher rates of physical and mental health comorbidities. As well as this, a greater number of prenatal substance exposures in addition to alcohol was found to be predictive of a child’s later involvement in the child welfare system, their own substance use when adjusted for their age, and the risk of developing physical and mental health comorbidities. These results suggest that prenatal exposure to other substances in addition to alcohol can significantly increase the severity of adverse physical health, mental health, and social outcomes, and highlight the synergistic effects of polysubstance exposure.

The children and youths in this study were diagnosed at an average age of 9.7 years, with the majority (53%) being 6 to 10 years old. Only 12% were aged 16 and above. This is somewhat older than the median age of 5.7 years reported in the 2019 Canadian Health Survey on Children and Youth [31]. Previous research has suggested that between 6 and 12 years is an opportune age for diagnosing FASD, as this is when sentinel facial features are most apparent and a full neurodevelopmental assessment is generally possible [30]. Though individuals of all ages can benefit from receiving an FASD assessment and accurate diagnosis, it is ideal if this occurs before the age of 18 so that appropriate supports can be put in place early and adverse health and social outcomes are minimized.

A substantial portion of the children and youths in this study (14%) had at least one sibling with FASD or a parent with suspected/confirmed FASD/PAE (15%), suggesting ongoing challenges with alcohol use within a subset of these families. The combination of prenatal exposure and living within an environment of substance use, particularly multi-substance use, could play a significant role in the development of later drug and alcohol problems in these children and youths [19,32]. The results from this study found a significant positive association between the number of prenatal substance exposures and the likelihood of a child’s substance use, adjusted for their age. Females with FASD may therefore be at an increased risk when they become sexually active of having alcohol-exposed pregnancies themselves, thus increasing the risk of FASD recurrence. This emphasizes the need to provide pre-conception prevention education and access to substance use treatment services to individuals with FASD to reduce the risk of FASD recurring in the family and to promote healthy, substance-free pregnancies. This might include providing access to programs specializing in alcohol use treatment and/or brief interventions for pregnant women. One Canadian example of this is the Alberta Parent–Child Assistance Program, which has been shown to be both effective and cost-effective [33,34,35] in reducing the risk of FASD recurrence and subsequent child welfare involvement.

The children and youths in this sample displayed high rates of physical and mental health comorbidities, and both FASD with SFF and increasing age were found to be significant in predicting increased comorbidities. The most common physical comorbidities were disorders of the respiratory system, disorders of the nervous system, and genetic/congenital problems, at rates of 10.1%, 9.4%, and 9.0%, respectively. Clinically, individuals with FASD frequently report challenges with pain, chronic conditions, and physical illnesses. The results from a recent survey of individuals with FASD also found high rates of many physical/medical conditions and concluded that FASD is best conceptualized as a “whole-body disorder” [36]. As well as physical challenges, the children and youths in this study also displayed many mental health comorbidities, including ADHD (66%), specific learning disorders (42%), language/speech delays or disorders (34%), intellectual disabilities (27%), and anxiety or panic disorders (21%). While highly reported, these numbers may not paint a full picture of the physical and mental health concerns that are commonly experienced by individuals with FASD across their lifespans. The fact that an early diagnosis of FASD was obtained for many individuals in this sample has presumably facilitated these children and youths to receive at least some FASD-informed care and services. Those diagnosed later in life may not have this advantage and may face even higher rates of adverse outcomes. Regardless of when an FASD diagnosis is received, it is likely that these individuals will require additional physical and mental health supports across their lifespan and will utilize healthcare services at a higher rate, incurring a high cost burden to service systems.

In this study, prenatal exposure to multiple substances in addition to alcohol was also common. These exposures included tobacco/cigarettes/nicotine (33.4%), cocaine/crack (27.0%), and cannabis (24.3%). Exposure to cocaine/crack (OR: 1.75), opioids (OR: 2.01), and polysubstance use (OR: 1.18) all significantly increased the likelihood of a diagnosis of FASD with SFF. It is important to note that data on prenatal medical conditions and substance exposures are limited to what was available in individual files and may not have been individually elicited or reported. As a result, these statistics likely underestimate maternal conditions and prenatal substance use during pregnancy.

Among children and youths in this study, 12.7% (*n* = 151) had been involved in the child welfare system either personally or through sibling involvement, and less than 2% had been involved with the justice system as either a perpetrator or a victim. Having a diagnosis of FASD with SFF was found to be associated with lower odds of both child welfare and criminal justice system involvement when adjusted for age, possibly due to having a more severe and identifiable developmental disability and possibly earlier diagnosis. This in turn could lead to more proactive, preventative support and the receipt of FASD-informed services, and therefore fewer interactions with justice or child protection systems.

The number of prenatal exposures in addition to alcohol, irrespective of the FASD diagnostic category, also significantly predicted involvement in the child welfare system (OR: 1.52), current substance use when adjusted for age (aOR: 1.51), and increases in mental health and physical comorbidities. This indicates that prenatal exposure to other substances in addition to alcohol can significantly increase the severity of FASD diagnoses and adverse health and social outcomes. It is therefore important to thoroughly screen women for all substances, especially when combined with alcohol. Service providers implementing FASD prevention strategies may choose to incorporate these substances in the design and implementation of brief interventions provided to women at risk of alcohol-exposed pregnancies. Additionally, access to pregnancy planning services, prenatal nutrition, and safe environments are important to facilitate healthy pregnancies, since women who experience violence from intimate partners [37] or women who have a history of mental health issues and substance use [19] are more likely to consume alcohol and other substances while pregnant.

This clinic-based cross-sectional study has several notable strengths, including the analysis of over 1000 patient records and the collection of detailed data on maternal experiences and child health/social outcomes from one of the two largest clinics in BC, Canada. This study also has its limitations, including the use of data based on self-reports, which may be influenced by stigma and social desirability bias and may therefore underestimate prenatal substance exposures, maternal experiences such as abuse or poverty, and the current substance use of the children and youths with FASD. As this study is cross-sectional and is based on data obtained at the time of diagnosis only, it does not necessarily portray the full range of adversities and successes that can be experienced by children and youths with FASD.

It is important to track these data over time to monitor children and youths who may be at risk of substance use or substance use disorders and other adverse health and social outcomes. Future studies can continue to explore the experiences of individuals living with FASD so as to advocate for and improve the accessibility of support services that would most benefit this population and ensure a quality of life that allows individuals with FASD to reach their full potential.

## Figures and Tables

**Table 1 nutrients-16-01655-t001:** Demographics of children and youths diagnosed with FASD (*n* = 1187).

Characteristics at Time of Assessment	Frequency	Percentage
Gender		
Male	729	61.4
Female	457	38.5
Year of diagnosis		
2015	365	30.7
2016	273	23.0
2017	301	25.4
2018	248	20.9
Age at diagnosis		
2 to 5 years	137	11.5
6 to 10 years	628	52.9
11 to 15 years	284	23.9
≥16 years	138	11.6
Range (years)	2–19	
Mean (SD)	9.7 (3.9)	
Diagnosis		
Fetal alcohol spectrum disorder without sentinel facial features	698	58.8
Fetal alcohol spectrum disorder with sentinel facial features	129	10.9
Fetal alcohol syndrome (with confirmed exposure)	S	S
Fetal alcohol syndrome (without confirmed exposure)	<5	S
Partial fetal alcohol syndrome (with confirmed exposure)	79	6.7
Alcohol-related neurodevelopmental disorder (with confirmed exposure)	269	22.7
Language(s) spoken at home		
English only	1163	98.0
English and one other language (not Aboriginal)	14	1.2
English and an Aboriginal language	6	0.5
Spanish, Portuguese, and English	<5	S
Aboriginal, English, and French	<5	S
Arabic	<5	S
Living arrangement (at the time of assessment)		
With biological parent(s)	428	36.1
In foster care	303	25.5
With extended family	218	18.4
In an adoptive home	188	15.8
In a group/resource home	13	1.1
With care provider	12	1.0
Independently	9	0.8
With guardian	6	0.5
With a step-parent	<5	S
With a significant other	<5	S
Unknown	6	0.5
Adopted		
Yes	200	16.8
N/A	987	83.2
Geographic location		
Urban	903	76.1
Rural	271	22.8
Unknown	13	1.1
Sibling(s) with FASD		
At least one sibling with FASD	161	13.6
Sibling(s) with suspected FASD/PAE	34	2.9
Sibling(s) with no maternal relation with FASD	<5	S
Parent(s) with suspected/confirmed FASD/PAE		
Yes	174	14.7
N/A	1013	85.3
Personal or sibling involvement in child welfare system		
Yes	151	12.7
N/A	1036	87.3
Personal justice system involvement		
Involvement as perpetrator	17	1.4
Involvement as victim	<5	S
N/A	S	S

N/A: not applicable; S: suppressed in order to prevent the identification of individuals. Please note that data on the languages spoken, justice system involvement, and child welfare system involvement are not consistently elicited or reported by clinicians in individual files.

**Table 2 nutrients-16-01655-t002:** Maternal conditions during pregnancy for mothers of children and youths diagnosed with FASD.

Maternal Conditions at Time of Assessment *	Frequency	Percentage
Physical medical condition ^a^	135	11.4%
Pregnancy complication	112	9.4%
No/poor/limited/irregular/delayed prenatal care	81	6.8%
Stress	58	4.9%
Abuse as a victim/perpetrator	54	4.5%
Mental disorder ^b^	40	3.4%
Intrauterine growth restriction	9	0.8%
Poor nutrition	8	0.7%
Homelessness	6	0.5%
Other/unspecified exposures	5	0.4%
Eating disorder	<5	S
Justice system involvement	<5	S
High-risk/chaotic lifestyle	<5	S
Occupational/environmental exposure	<5	S

S: suppressed in order to prevent the identification of individuals. * Categories are not mutually exclusive. ^a^ Includes women with indication of the following: surgical procedure, injury, physical health issue, or undefined medical condition. ^b^ Includes women with indications of symptoms/factors impacting their mental health and excludes women taking prescription drugs for mental health disorders without any indication of the disorder.

**Table 3 nutrients-16-01655-t003:** Prenatal exposure of children and youths diagnosed with FASD to substances.

Confirmed/Suspected Prenatal Exposures *	Frequency	Percentage
Cigarettes/tobacco/nicotine	396	33.4%
Cocaine/crack	321	27.0%
Marijuana/cannabis	288	24.3%
Unspecified drug/substance (illegal and legal)	124	10.4%
Opioid/opiate/heroin (excluding methadone maintenance treatment)	121	10.2%
Stimulant drug	113	9.5%
Other medication/supplement	89	7.5%
Antidepressants	34	2.9%
Benzodiazepine	26	2.2%
Antipsychotic/antimanic medication	23	1.9%
Hallucinogen	8	0.7%
Polysubstance **	6	0.5%
Unspecified drug for mental illness/drug for anxiety	6	0.5%
Overdose	5	0.4%
Methadone maintenance treatment	5	0.4%
Rehab/clinic admission	<5	S
Caffeine	<5	S
Secondhand exposure to tobacco	<5	S
Secondhand exposure to marijuana	<5	S

S: suppressed in order to prevent the identification of individuals. * Categories are not mutually exclusive. ** Included observations where polysubstance use was explicitly stated.

**Table 4 nutrients-16-01655-t004:** Physical and mental health comorbidities of children and youths diagnosed with FASD.

Comorbidities *	Frequency	Percentage
** *Physical health comorbidities* **		
Respiratory system disorders	120	10.1%
Nervous system disorders	112	9.4%
Genetic problems/congenital deformation/malformation	107	9.0%
Eye-related conditions	87	7.3%
Ear-related conditions	57	4.8%
Cardiovascular system disorders	33	2.8%
Digestive system disorders	32	2.7%
Connective tissue/musculoskeletal disorders	32	2.7%
Nutrition/metabolism/endocrine system disorders	23	1.9%
Blood/immune response disorders	22	1.9%
Injuries/harmful exposures to lead/anaphylaxis	18	1.5%
Genitourinary system disorders	17	1.4%
Infectious diseases	14	1.2%
Skin conditions	13	1.1%
Problem starting within the perinatal period ^a^	11	0.9%
Other factors impacting health	11	0.9%
Neonatal abstinence syndrome	8	0.7%
Procedure/brace/cast ^b^	8	0.7%
Overdose/intoxication/substance use	6	0.5%
Visual/auditory hallucinations, unspecified	5	0.4%
Tumor	<5	S
Self-injury/self-injury ideation	<5	S
** *Mental health comorbidities* **		
ADHD	789	66.5%
Specific learning disorders (reading, writing, mathematic or scholastic skills)	497	41.9%
Language/speech disorders or delays (expressive or receptive)	400	33.7%
Intellectual disabilities	317	26.7%
Anxiety (social; separation; generalized; unspecified) or panic disorders	254	21.4%
Motor, tic, or developmental coordination disorders	114	9.6%
Oppositional conduct or defiant/disruptive behavior/disorders	101	8.5%
Adjustment or attachment disorders	108	9.1%
PTSD	89	7.5%
Developmental delay (global and unspecified)	79	6.7%
Depressive disorders	60	5.1%
Autism	30	2.5%
Sleep disorders	21	1.8%
Substance use disorders	17	1.4%

* Categories are not mutually exclusive; each child/youth may have more than one comorbidity. ^a^ Includes neonatal stroke, congenital hypotonia, brachial plexus injury, birth palsy, cerebral ischemic or hypoxic ischemic injury, birth asphyxia (brain injury), bronchopulmonary dysplasia, amniotic band syndrome, prematurity, macrosomia, and fetuses and newborns affected by premature rupture of membranes. ^b^ Observations where the comorbidity requiring the procedure/treatment was undefined.

**Table 5 nutrients-16-01655-t005:** Odds estimates for the diagnosis of FASD with sentinel facial features based on prenatal exposures in addition to alcohol among children and youths.

Exposure Variable	Odds of FASD w SFF (95% CI)	*p*-Value
Prenatal cigarette/nicotine/tobacco exposure	1.41 (0.99, 2.03)	0.059
Prenatal marijuana exposure	0.95 (0.63, 1.43)	0.800
Prenatal cocaine/crack exposure	1.75 (1.21, 2.53)	0.003 *
Prenatal opioid exposure	2.01 (1.23, 3.26)	0.005 *
Limited prenatal care	0.92 (0.45, 1.89)	0.825
Maternal mental health issues	1.60 (0.69, 3.70)	0.268
Poor nutrition	1.06 (0.13, 8.68)	0.957
Number of prenatal substances (in addition to alcohol)	1.18 (1.04, 1.33)	0.009 *

SFF: sentinel facial features. *—statistically significant at the 0.05 alpha level.

**Table 6 nutrients-16-01655-t006:** Odds ratios and parameters of linear regression models for social and health outcomes based on the diagnosis of FASD with sentinel facial features among children and youths.

Outcome Variable	OR (95% CI)	*p*-Value	OR (95% CI) ^a^	*p*-Value ^a^	OR (95% CI) ^b^	*p*-Value ^b^
Child welfare involvement	1.00 (0.59, 1.70)	0.986	0.97 (0.57, 1.65)	0.908	0.87 (0.50, 1.50)	0.611
Criminal justice system involvement	0.41 (0.05, 3.08)	0.385	0.78 (0.09, 6.15)	0.815	0.78 (0.10, 6.21)	0.818
Current substance use	0.19 (0.04, 0.80)	0.024 *	0.49 (0.10, 2.39)	0.385	0.51 (0.10, 2.52)	0.409
	R^2^, F-test	*p*-value	R^2^, F-test ^a^	*p*-value ^a^	R^2^, F-test ^b^	*p*-value ^b^
Number of mental health comorbidities	R^2^ = 0.0002, F (1,1185) = 0.19	0.664	R^2^ = 0.0059, F (2,1185) = 3.51	0.030 *	R^2^ = 0.023, F (3,1185) = 9.51	0.000 *
Number of physical comorbidities	R^2^ = 0.001, F (1,1185) = 19.27	0.000 *	R^2^ = 0.019, F (2,1184) = 11.47	0.000 *	R^2^ = 0.071, F (3,1183) = 30.51	0.000 *

OR: odds ratio; ^a^—adjusted for age; ^b^—adjusted for age and the number of prenatal substances. *—statistically significant at the 0.05 alpha level.

**Table 7 nutrients-16-01655-t007:** Odds ratios and parameters of linear regression models for social and health outcomes based on prenatal substance exposures, in addition to alcohol exposure, among children and youths.

Outcome Variable	OR (95% CI)	*p*-Value	OR (95% CI)^a^	*p*-Value ^a^	OR (95% CI) ^b^	*p*-Value ^b^
Child welfare involvement	1.52 (1.35, 1.71)	0.000 *	1.53 (1.35, 1.72)	0.000 *	1.53 (1.35, 1.72)	0.000 *
Criminal justice system involvement	1.05 (0.76, 1.46)	0.771	1.33 (0.94, 1.89)	0.106	1.33 (0.94, 1.88)	0.106
Current substance use	0.94 (0.79, 1.13)	0.539	1.51 (1.18, 1.93)	0.001 *	1.51 (1.18, 1.93)	0.001 *
	R^2^, F-test	*p*-value	R^2^, F-test ^a^	*p*-value ^a^	R^2^, F-test ^b^	*p*-value ^b^
Number of mental health comorbidities	R^2^ = 0.0140, F (1,1185) = 16.79	0.000 *	R^2^ = 0.0234, F (2,1184) = 14.17	0.000 *	R^2^ = 0.0235, F (3,1183) = 9.51	0.000 *
Number of physical comorbidities	R^2^ = 0.0599, F (1,1185) = 75.52	0.000 *	R^2^ = 0.0605, F (2,1184) = 38.11	0.000 *	R^2^ = 0.0718, F (3,1183) = 30.51	0.000 *

OR: odds ratio; ^a^—adjusted for age; ^b^—adjusted for age and the FASD diagnostic category (with or without SFF). *—statistically significant at the 0.05 alpha level.

## Data Availability

Restrictions apply to the availability of these data. Data were obtained from the Sunny Hill Health Centre and are available from Dr. Svetlana Popova with the permission of the Sunny Hill Health Centre and the Centre for Addiction and Mental Health.

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
