# Peer review of "Prenatal Exposures, Diagnostic Outcomes, and Life Experiences of Children and Youths with Fetal Alcohol Spectrum Disorder"

_nutrients, 2024, doi:10.3390/nu16111655_

Round 1

Reviewer 1 Report

Comments and Suggestions for Authors

This is a sound study on FASD plus prenatal drug exposition. Surprisingly, FASD with SFFwas significantly more likely to be diagnosed following prenatal exposure to drugs. I think, this results needs to be discussed in (much) more detail as substance use in pregnancy may imply a reduction of PAE.

I think that a major flaw of the study is the lack of information on the cognitive development of the subgroups. Studies show that children prenatally exposed to alcohol and drugs have a higher FSIQ than children exposed solely to alcohol. Unfortunately, patient charts rarely include cognitive test results.

Author Response

Manuscript ID: nutrients-3007054

Title: Prenatal exposures, diagnostic outcomes and life experiences of children and youth with Fetal Alcohol Spectrum Disorder

Authors: Svetlana Popova*, Danijela Dozet, Mary-Rose Faulkner, Lesley Howie, Valerie Temple

Responses to Reviewers

Reviewer 1.

  1. This is a sound study on FASD plus prenatal drug exposition. Surprisingly, FASD with SFF was significantly more likely to be diagnosed following prenatal exposure to drugs. I think, this results needs to be discussed in (much) more detail as substance use in pregnancy may imply a reduction of PAE.

Response:

Unfortunately, we cannot ascertain the quantity or frequency of prenatal alcohol exposure in this population of children and youth with FASD. Prenatal alcohol exposure is assumed or reported based on the diagnosis of FASD that has been provisioned for each child. As well, prenatal exposure to substances, in addition to alcohol, is based on self-report, which may be subject to social desirability bias and thus may be underreported. Therefore, we hesitate to discuss the results of alcohol use frequency and quantity as it relates to use of other substances during pregnancy in terms of exact usage by the pregnant woman. However, we have indicated that use of prenatal substances has the potential to worsen the adverse impacts of prenatal alcohol exposure.

  1. I think that a major flaw of the study is the lack of information on the cognitive development of the subgroups. Studies show that children prenatally exposed to alcohol and drugs have a higher FSIQ than children exposed solely to alcohol. Unfortunately, patient charts rarely include cognitive test results.

Response:

Unfortunately, the data on the cognitive development of the subgroups are not available in this dataset.

Reviewer 2 Report

Comments and Suggestions for Authors

The article presents the findings of a study on the health and social outcomes of children and young people diagnosed with fetal alcohol spectrum disorder (FASD) in British Columbia, Canada. A comprehensive review of the literature on this topic reveals that numerous publications have already been published on this subject. Therefore, it is recommended that the authors focus on highlighting the novel aspects of their study compared to previous research, and only report on the findings that are unique to this survey and how they differ from those of previous studies. Moreover, It would be beneficial to provide an overview of the resources currently available to individuals with FASD and their families, and to assess the potential contribution of this work in terms of advancing knowledge and practice beyond what is currently known and done.

Please update the bibliography by adding the following paper:

doi: 10.1016/j.drugalcdep.2020.108487

doi: 10.1111/cch.12817.

doi: 10.1016/j.ridd.2019.103516.

doi: 10.1136/bmjopen-2022-065005.

Minor request:

·       “Study Design and Data Source” paragraph should be shortened and combined with the “Study population” paragraph.

·       Please correct the Article structure follow the author guidelines.

Author Response

Manuscript ID: nutrients-3007054

Title: Prenatal exposures, diagnostic outcomes and life experiences of children and youth with Fetal Alcohol Spectrum Disorder

Authors: Svetlana Popova*, Danijela Dozet, Mary-Rose Faulkner, Lesley Howie, Valerie Temple

Responses to Reviewers

Reviewer 2.

  1. The article presents the findings of a study on the health and social outcomes of children and young people diagnosed with fetal alcohol spectrum disorder (FASD) in British Columbia, Canada. A comprehensive review of the literature on this topic reveals that numerous publications have already been published on this subject. Therefore, it is recommended that the authors focus on highlighting the novel aspects of their study compared to previous research, and only report on the findings that are unique to this survey and how they differ from those of previous studies.

Response:

This study is unique, as it is the first study, which used patient chart data housed at the Sunny Hill Centre for Children in BC, Canada. This Centre is part of the Complex Developmental Behavioural Conditions (CDBC) network, which provides diagnostic assessments across whole province for children and youth ages 18 months to 19 years. All findings in this study are novel (e.g., FASD diagnostic outcomes based on prenatal exposures and health/social outcomes based on FASD diagnoses). We believe that we sufficiently supported the discussion of our findings with references of existing research studies.

Moreover, it would be beneficial to provide an overview of the resources currently available to individuals with FASD and their families, and to assess the potential contribution of this work in terms of advancing knowledge and practice beyond what is currently known and done.

Response:

A respective reference for available support resources was added.

  1. Please update the bibliography by adding the following paper:

doi: 10.1016/j.drugalcdep.2020.108487 - added

doi: 10.1111/cch.12817. – We do not feel that this study fits with our findings or discussion in general. We did not examine the impacts of child welfare system involvement.

doi: 10.1016/j.ridd.2019.103516. - added

doi: 10.1136/bmjopen-2022-065005. - added

  1. Minor request: “Study Design and Data Source” paragraph should be shortened and combined with the “Study population” paragraph.

Response:

We revised and shortened “Study population”. We prefer to keep these sections separately.

  1. Please correct the Article structure follow the author guidelines.

Response:

The paper is now formatted correctly as per the author guidelines.